# Parameters Optimization and Performance Evaluation Model of Air-Assisted Electrostatic Sprayer for Citrus Orchards

**Xiuyun Xue** [1,2,3,4], **Kaixiang Zeng** [1], **Nengchao Li** [1], **Qin Luo** [1], **Yihang Ji** [1], **Zhen Li** [1,2,3,4,*], **Shilei Lyu** [1,2,3,4] **and Shuran Song** [1,2]

1   College of Electronic Engineering (College of Artificial Intelligence), South China Agricultural University, Guangzhou 510642, China; xuexiuyun@scau.edu.cn (X.X.); lvshilei@scau.edu.cn (S.L.); songshuran@scau.edu.cn (S.S.)
2   National Citrus Industry Technical System Machinery Research Office, Guangzhou 510642, China
3   Guangdong Provincial Agricultural Information Monitoring Engineering Technology Research Center, Guangzhou 510642, China
4   Key Laboratory of Smart Agricultural Technology in Tropical South China, Ministry of Agriculture and Rural Affairs, Guangzhou 510642, China
*   Correspondence: lizhen@scau.edu.cn

**Abstract:** Citrus orchards in Southeast Asia are commonly grown in hilly areas, where the terrain is unsuitable for the operation of crop protection machinery. Conventional spraying equipment used in hilly orchards have a poor deposition effect. In this paper, a new air-assisted electrostatic sprayer was designed for hilly citrus orchards. The orthogonal method was conducted to determine the optimal spray parameters of the sprayer. To evaluate the spray performance of the optimized air-assisted electrostatic sprayer, field tests were carried out on a citrus orchard with various cultivation patterns. Based on the data of the field tests, a comprehensive evaluation model was constructed to quantitatively analyze the performance of the sprayer. Results indicate that the optimal parameters are a spray pressure of 0.5 MPa, applied voltage of 9 kV and air flow velocity of 10 m/s. The optimized air-assisted electrostatic sprayer has the best performance in the citrus under dense fence cultivation pattern, followed by dense dwarf cultivation pattern. Comparing to the other sprayers tested, the air-assisted electrostatic sprayer greatly improves the spray coverage on the leaf surfaces (abaxial and adaxial) under various cultivation patterns.

**Keywords:** air-assisted electrostatic sprayer; parameters optimization; spray performance; evaluation model

## 1. Introduction

Citrus fruits are some of the most important fruits in the global fruit industry, planted in about 114 countries, with an annual global yield of approximately 140 million tons [1,2]. Crop protection is a key process in orchard management, which can ensure the yield and quality of fruit production. However, in Southeast Asia, the majority of citrus are grown in hilly orchards with complex terrain [3]. It is hard for crop protection machinery to operate in such areas. At present, hilly orchards mostly use hand-pressure-based sprayers, hand-held motorized sprayers and hand-held spray guns, as these are low-cost and adapt well to complex terrain [4]. However, such kinds of equipment have delivered excessive pesticide with low spray deposition efficiency and high pesticide drift loss, which poses a serious problem of environmental contamination [5–7].

Air-assisted electrostatic spray technology is a combination of air-assisted spray technology and electrostatic spray technology. This technology employs a high voltage to charge the spray liquid, establishing an induced electric field between the nozzle and the canopy [8]. The charged droplets mutually repulse one another and deposit uniformly on the leaf surfaces under the force of the electric field and airflow [9]. Spray pressure,

applied voltage and air flow velocity are the main factors which influence the spray performance of an air-assisted electrostatic sprayer [10,11]. Finding the optimal combination of factors/levels for the air-assisted electrostatic sprayer is the most important step in the sprayer design to minimize pesticide use and improve spray efficiency.

The airflow generated by the air-assisted electrostatic sprayer should be strong enough to move the leaves in the canopy. According to the principle of air-assisted spraying, the volume of airflow should be equal to the volume of air replaced within the canopy [12]. This is greatly dependent on the width and spatial construction of the canopy [13,14]. Cultivation pattern is an important factor in regulating the spatial construction of the canopy. Currently, the majority of hilly citrus cultivation adopt general cultivation patterns. Most citrus orchards in such a pattern are bush trees with a high canopy density. Semi-standardization hilly orchards commonly use the dense dwarf cultivation pattern. In some regions, the exploration of a dense fence cultivation pattern is also being conducted [15]. Such dense patterns are beneficial to improve the internal spatial construction of the canopy, reducing the canopy density. However, there is less work to evaluate the spray performance of an air-assisted electrostatic sprayer in citrus under various cultivation patterns.

Water consumption, spray coverage and spray distribution are important indexes which evaluate the spray performance of sprayers [16,17]. In existing studies, the evaluation of spray performance is commonly analyzed by using one or two evaluation indexes under different operating conditions, thus lacking a quantitative analysis of comprehensive indexes [18]. Such a method cannot evaluate the performance of the sprayer overall. Therefore, to quantitatively analyze the spray performance of the air-assisted electrostatic sprayer, it is necessary to construct a comprehensive evaluation model.

In this paper, a new air-assisted electrostatic sprayer is designed to address the problems of conventional sprayers of low spray deposition efficiency, non-uniform spray distribution within the canopy and significant losses to the ground in hilly citrus orchards. To obtain the optimal spray parameters, the orthogonal method is performed based on spray pressure, applied voltage and air flow velocity at the air outlet as the factors. The performance of the optimized air-assisted electrostatic sprayer is evaluated in a citrus orchard with various cultivation patterns. Furthermore, the comprehensive evaluation models were constructed to quantitatively analyze the spray performance of the sprayers.

## 2. Materials and Methods

### 2.1. Air-Assisted Electrostatic Sprayer Design

As shown in Figure 1, the air-assisted electrostatic sprayer (AAES) mainly consists of a water pump, water tank, high-voltage electrostatic generator, adjustable blower and electrostatic nozzle, etc. The electrostatic nozzle (spray pressure variation range: 0–0.7 MPa) is composed of a solid-cone nozzle and a ring electrode. The nozzle charges conductive liquid and droplets based on the induction principle, which is the most reliable and safest method applied to portable sprayers [19,20]. The water pump (flow variation range: 0–5 L/min) supplies liquid for electrostatic nozzle and is powered by a 12 V portable power source. The high-voltage electrostatic generator (model no: JDFS-01; size: $120 \times 28 \times 25$ mm; input: 12 V; output voltage variation range: 0–10 kV) is connected with the ring electrode and provides high-voltage static electricity up to 10 kV. The adjustable blower is embedded in the sprayer so that the nozzle can keep dry and prevent the occurrence of electrical leakage. By adjusting the power of the blower, the air flow velocity at the air outlet of AAES can be varied from 0 to 10 m/s.

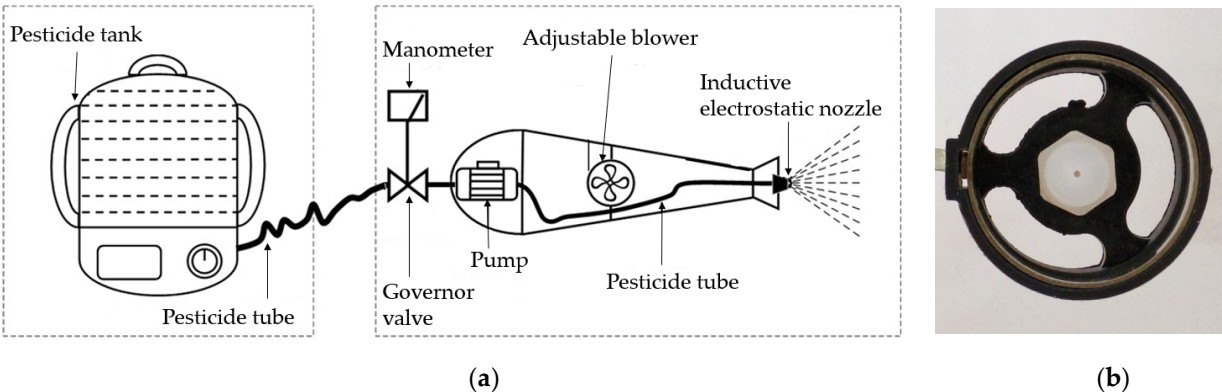

**Figure 1.** (**a**) The structure diagram of AAES. (**b**) The electrostatic nozzle.

*2.2. Orthogonal Experiment Design*

To study the effects of different spray parameters on the spray performance, determining the optimal spray parameters combination, an $L_9$ ($3^4$) orthogonal table is carried out based on three factors: spray pressure (A), applied voltage (B) and air flow velocity (C). The factors and levels of the orthogonal test are shown in Table 1. In order to evaluate the spray performance of the sprayer correctly, charge-to-mass ratio (CMR), volume median diameter (VMD), relative span (RS) and spray coverage are selected as the evaluation indexes. Thereof, CMR is used to evaluate the charging performance; VMD and RS are used to evaluate the droplet size together; and spray coverage is used to evaluate the droplets deposition performance. Whereafter, the range analysis of the test results is carried out to select the optimal spray parameters combination.

**Table 1.** Orthogonal experiment table.

| Test Number | Factor | | |
| :---: | :---: | :---: | :---: |
| | Spray Pressure A/(MPa) | Applied Voltage B/(kV) | Air Flow Velocity C/(m/s) |
| 1 | 0.3 | 3 | 6 |
| 2 | 0.3 | 6 | 8 |
| 3 | 0.3 | 9 | 10 |
| 4 | 0.5 | 3 | 8 |
| 5 | 0.5 | 6 | 10 |
| 6 | 0.5 | 9 | 6 |
| 7 | 0.7 | 3 | 10 |
| 8 | 0.7 | 6 | 6 |
| 9 | 0.7 | 9 | 8 |

*2.3. Measurement of CMR*

The charge-to-mass ratio (CMR) is one of the most important indexes to reflect the performance of the electrostatic sprayer. A higher CMR leads to an increase in the force of attraction between the charged droplets and the canopy [9], which results in a higher spray coverage and droplets deposition uniformity. To measure CMR, the Faraday cylinder method has been used for the experiment [21,22]. As shown in Figure 2, the measurement set-up includes a Faraday cylinder (diameter: 45 cm; height: 50 cm), a digital multimeter (model no: HEST-112A;) and electronic scales. The Faraday cylinder is connected to the earth potential via digital multimeter and is insulated from the ground.

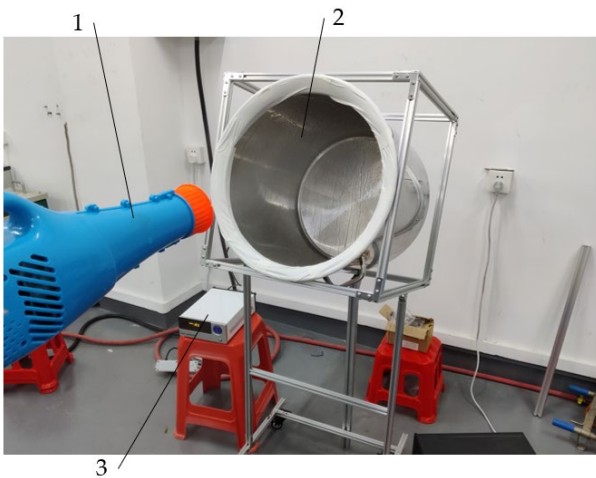

**Figure 2.** Experimental set-up to measure CMR. 1. Sprayer. 2. Faraday cylinder. 3. digital multimeter.

During the measurement, it is preferable to position the nozzle as close as possible to the center of the Faraday cylinder so that the droplets could be collected as much as possible. The contact of the charge droplets on to the Faraday cylinder will generate an electrical current, measuring the current and weight of the charge droplets within a specific time. The CMR is calculated from Equation (1).

$$\text{CMR} = \frac{i_s}{Q_m} \tag{1}$$

where $i_s$ is the measured spray current, $Q_m$ is the mass of liquid.

### 2.4. Measurement of the Droplet Size

Phase Doppler Anemometry (PDA) is an extension of Laser Doppler Anemometry (LDA) and finds widespread application in various fields such as fluid dynamics, combustion and spray analysis. PDA relies on the principles of interferometry and the Doppler effect to measure the properties of flowing particles or droplets in a fluid medium [23,24]. As shown in Figure 3, the PDA system consists of a laser source, a pair of detectors and a coordinate rack. During the measurement, the laser beam passes through the charged aerosol at a distance of 1 m from the nozzle.

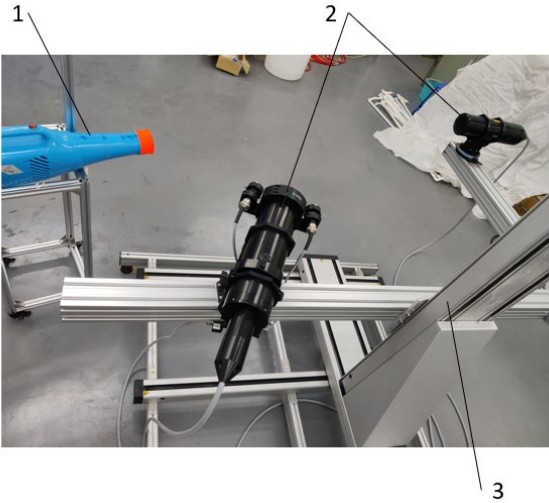

**Figure 3.** Experimental set-up to measure the droplets size. 1. Sprayer. 2. The detectors of PDA. 3. The coordinate rack of PDA.

By analyzing the Doppler-equivalent frequency of the laser light scattered by the droplets within the spray, PDA enables simultaneous measurement of droplet size and distribution indexes such as $D_{V10}$, VMD, $D_{V90}$, RS, etc. VMD refers to the droplet size where the cumulative volume of droplets equal to or smaller than that size accounts for 50% of the total volume [25]. RS provides a measurement of the uniformity of the particle size distribution relative to the median particle size [26]. The smaller the RS, the more uniform the droplets size distribution. The VMD and RS are calculated following Equations (2) and (3).

$$\text{VMD} = \left( \frac{\sum d_i^3 N_i}{\sum N_i} \right)^{\frac{1}{3}} \tag{2}$$

$$\text{RS} = \frac{D_{V90} - D_{V10}}{D_{V50}} \tag{3}$$

### 2.5. Measurement of the Spray Coverage

The spray coverage measurement was performed in the laboratory with an artificial citrus (tree high: 200 cm; canopy width: 167 cm). The artificial citrus could ensure that the indicator can be measured under controlled environment conditions [27].

In the measurement, the canopy was divided into the top, middle and bottom layers (Figure 4). There were three sampling points in each layer, labeled as A, O and B, with a total of nine sampling points in the canopy. The sampling points labeled as O were set in the center of each layer. Thus, the canopy was divided into nine areas. At each sampling point, water-sensitive papers were placed on both the adaxial (upper side) and abaxial (underside) leaf surfaces to measure the spray coverage. The nozzle was positioned at a distance of about 1 m from the canopy to spray at a forward speed of 0.5 m/s. The water-sensitive papers were collected immediately after spraying and gathered in sealed bags until the droplets dried. The spray coverage was analyzed by the public domain software ImageJ (ImageJ 1.53, Wayne Rasband, National Institutes of Health, Bethesda, MD, USA).

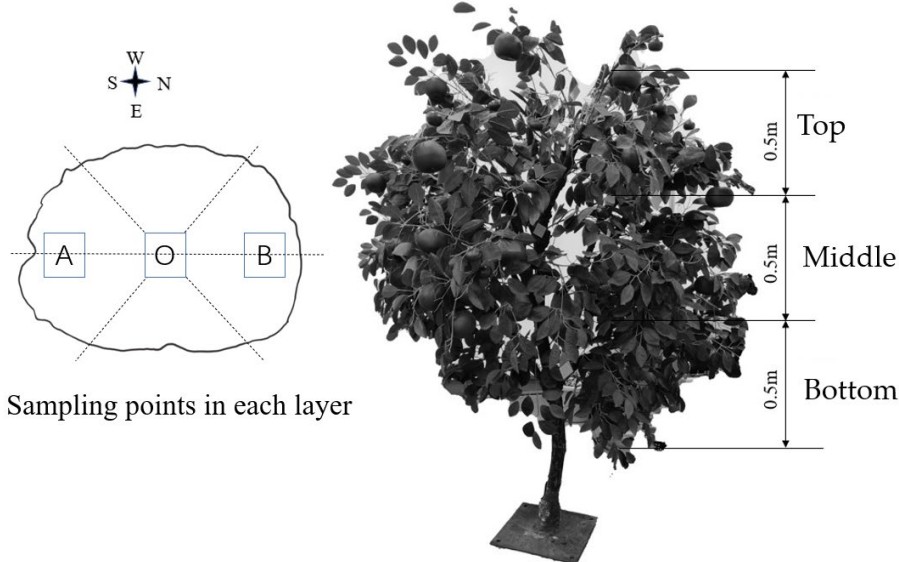

**Figure 4.** Design of sampling points for a citrus canopy.

### 2.6. Field Tests in a Citrus Orchard with Different Cultivation Patterns

In order to verify the spray performance of the optimized AAES, field tests were performed on citrus in different cultivation patterns according to the optimal spray parameters. In the tests, one electric sprayer (ES) and one spray gun (SG) were evaluated and compared with AAES. ES had the same design as AAES but without air-charge-assistance. Before

the field tests, flow rate, spray angle of water discharged from nozzle, maximum spray distance and water consumption of each sprayer were accurately measured. Spray pattern profiles discharged from the nozzle were determined from images taken with a high-speed camera (model no.: PCO pco.dimax csl) and the spray angles were measured by image analysis [28]. The specific spray application parameters of the sprayers under 0.5 MPa spray pressure are shown in Table 2.

**Table 2.** Spray application parameters for the test.

| Spray Application Parameters | AAES | ES | SG |
|---|---|---|---|
| Flow rate (L·min$^{-1}$) | 1.59 | 1.59 | 2.36 |
| Spray angle (Degrees) | 64.8 | 55.5 | 55.9 |
| Maximum spray distance (m) | 3.7 | 1.9 | 1.6 |
| Water consumption (L·ha$^{-1}$) | 406 | 954 | 1665 |

AAES: air-assisted electrostatic sprayer, ES: electric sprayer, SG: spray gun.

The field tests were carried out in a citrus orchard located in Ganzhou, Jiangxi Province, China. Citrus in three cultivation patterns were planted in different areas of the orchard, i.e., general pattern (P1), dense dwarf pattern (P2) and dense fence pattern (P3) (Figure 5). Table 3 showed the planting parameters of different cultivation patterns. Three typical citrus orchards in each cultivation pattern were selected and sampled. The method of sampling was the same as that in the laboratory test (Figure 4). During the field tests, the operating distance under each pattern was about 1 m. The operator moved forward at a constant speed of about 0.5 m/s to avoid excessive spraying. To characterize the uniformity of the spray distribution on the canopy, the coefficient of variation (CV) of spray coverage between layers was calculated using Equations (4) and (5).

$$S = \sqrt{\frac{\sum\limits_{i=1}^{3}\left(X_i - \overline{X}\right)^2}{3-1}} \tag{4}$$

$$CV = \frac{S}{\overline{X}} \times 100\% \tag{5}$$

where $X_i$ is the spray coverage on the adaxial/abaxial leaf surfaces in each layer, and $\overline{X_i}$ is the mean value of the spray coverage on adaxial/abaxial leaf surfaces in different layers.

**Table 3.** Planting parameters of citrus under different cultivation patterns.

| Cultivation Pattern | Row Spacing (m) | Plant Spacing (m) | Canopy Width (m) | Tree Height (m) |
|---|---|---|---|---|
| General | 3 | 1.3 | $1.7 \pm 0.4$ | $1.8 \pm 0.4$ |
| Dense dwarf | 4 | 1.5 | $1.4 \pm 0.2$ | $1.1 \pm 0.3$ |
| Dense fence | 4.5 | 1 | $0.7 \pm 0.3$ | $1.9 \pm 0.2$ |

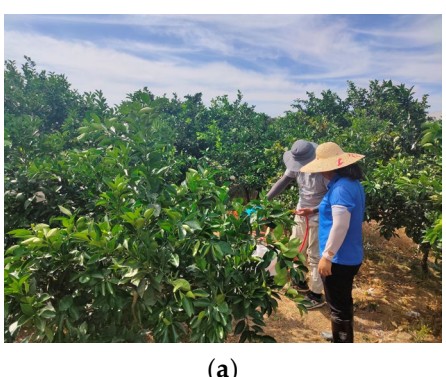 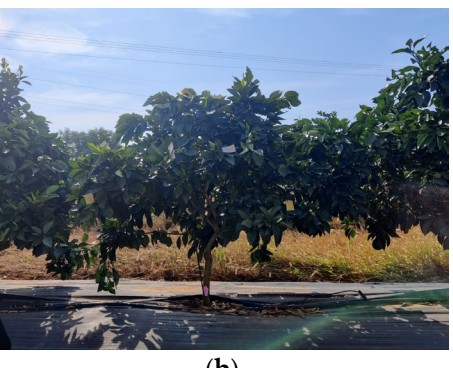 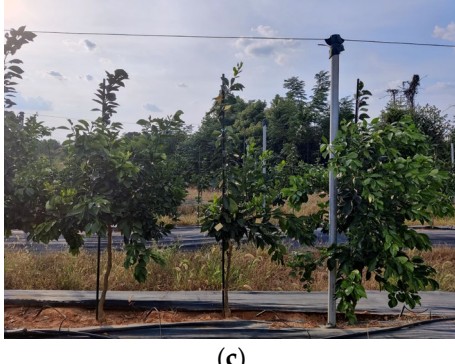

(**a**)                            (**b**)                            (**c**)

**Figure 5.** (**a**) Citrus orchards under the general cultivation pattern; (**b**) Citrus orchards under the dense dwarf cultivation pattern; (**c**) Citrus orchards under the dense fence cultivation pattern.

*2.7. Spray Performance Comprehensive Evaluation Model*

Based on the field tests data, a comprehensive evaluation model was constructed, which could accurately reflect the spray performance of the sprayer from various aspects. The model selected water consumption, spray coverage and the coefficient of variation (CV) of spray coverage between layers as the comprehensive evaluation indexes for spray performance. The steps of the comprehensive evaluation model construction are as follows:

1.  Indexes conversion; In the comprehensive evaluation model, all the indexes should be dimensionless. Here, the equalization method was used to convert the indexes to dimensionless indexes. In addition, negative indexes (i.e., water consumption, CV) were converted to positive indicators using the reciprocal method.

2.  Weight determination; The subjective weight method (SWM) and objective weight method (OWM) were used to determine the weight of the indexes [29]. Here, the subjective weight values were 1/5 and the values of objective weight were calculated based on the CV of the comprehensive evaluation indexes, as shown in Equations (6) and (7).

$$C_j = \frac{\left[ \frac{1}{3} \sum_{k=1}^{3} \left( x_{kj} - \overline{x_j} \right)^2 \right]^{\frac{1}{2}}}{\overline{x_j}} \tag{6}$$

$$\omega_j = \frac{C_j}{\sum_{j=1}^{5} C_j} \tag{7}$$

where $C_j$ is the coefficient of variation of index $j$, $x_{kj}$ refers to the field tests data of the three sprayers under different cultivation patterns, $\overline{x_j}$ is the mean value of index $j$ for three sprayers, and $\omega_j$ is the weight of index $j$.

3.  Model construction; The multi-indicator comprehensive evaluation model usually includes a linear weighted model or nonlinear weighted model. In this paper, the linear weighted method was used to construct the model.

## 3. Results and Discussion

*3.1. Analysis of the Orthogonal Experiment and Optimal Level of Spray Parameters*

Range analysis can evaluate the importance of each spray parameter through the values of R and $\overline{K_i}$, thus finding the optimal combination of parameters [30]. The range analysis results of the orthogonal experiment are shown in Table 4. The influence order of the principal factors on different indexes could be concluded as follows: B > C > A for CMR, A > C > B for VMD, C > A > B for RS and A > B > C for spray coverage.

Table 5 shows the optimal level of four indexes. The orthogonal experiment in this paper was multi-index. Thus, the optimal level should be determined by the optimization

objectives as well as the analysis results of indexes. The optimization of spray parameters should improve the spray coverage and spray distribution as much as possible. Spray pressure had significant influence on VMD and spray coverage. The results show that factor A in level 3 would have a finer droplets size, while factor A in level 1 would have a coarser droplets size. Fine droplets are prone to drift, while coarse droplets are not conducive to droplets deposition, hence reducing droplets penetration. Thus, the droplet size should be at an appropriate level. Factor A in level 2 has a medium droplets size, which reduces droplet drift, on the one hand, and has better spray coverage on the other hand. Therefore, A in level 2 would be considered the most suitable level. The optimal spray parameters combination for AAES is determined as $A_2B_3C_3$, which corresponds to a spray pressure of 0.5 MPa, applied voltage of 9 kV and air flow velocity of 10 m/s.

**Table 4.** Range analysis table.

| Index | Value | Spray Pressure A/(MPa) | Applied Voltage B/(kV) | Air Flow Velocity C/(m/s) |
|---|---|---|---|---|
| CMR $(mC \cdot kg^{-1})$ | $\overline{K_1}$ | 0.169 | 0.071 | 0.165 |
| | $\overline{K_2}$ | 0.168 | 0.186 | 0.142 |
| | $\overline{K_3}$ | 0.146 | 0.224 | 0.174 |
| | $R$ | 0.023 | 0.153 | 0.032 |
| VMD $(\mu m)$ | $\overline{K_1}$ | 240.23 | 218.37 | 221.00 |
| | $\overline{K_2}$ | 208.54 | 215.07 | 229.88 |
| | $\overline{K_3}$ | 196.74 | 212.07 | 194.63 |
| | $R$ | 43.49 | 6.30 | 35.26 |
| RS | $\overline{K_1}$ | 0.85 | 0.88 | 0.79 |
| | $\overline{K_2}$ | 0.85 | 0.86 | 0.85 |
| | $\overline{K_3}$ | 0.90 | 0.85 | 0.96 |
| | $R$ | 0.05 | 0.03 | 0.17 |
| spray coverage (%) | $\overline{K_1}$ | 11.46 | 12.58 | 13.43 |
| | $\overline{K_2}$ | 14.04 | 14.5 | 13.96 |
| | $\overline{K_3}$ | 17.36 | 15.72 | 15.48 |
| | $R$ | 5.90 | 3.14 | 2.05 |

**Table 5.** The optimal combination of the orthogonal experiment.

| Analysis Method | CMR | VMD | RS | Spray Coverage |
|---|---|---|---|---|
| Range analysis | $A_1B_3C_3$ | $A_3B_3C_3$ | $A_1B_3C_1$ | $A_3B_3C_3$ |

*3.2. Analysis of Field Tests*

3.2.1. Spray Coverage for AAES under Different Cultivation Patterns

The spray coverage on leaf surfaces (abaxial and adaxial) under three cultivation patterns for AAES are reported in Table 6. The order of mean spray coverage on adaxial leaf surfaces under three cultivation patterns was as follows: P3 (Dense fence) > P2 (Dense dwarf) > P1 (General). In comparison with P1, mean coverage on adaxial leaf surfaces under P2 and P3 increased by 9.4% and 11.79%, respectively. This significant differences were mainly due to a significantly lower spray coverage in the middle layers as well as bottom layers under P1. Compared to P2, the mean spray coverage in the middle layers and bottom layers in P1 decreased by 9.51% and 21.18%, and compared to P3, it decreased by 10.68% and 20.91%, respectively. In addition, P1 had less uniformity of spray distribution in the canopy (CV between layers (adaxial) of 56.64%) compared to the other patterns. Such coverage trends can be explained based on the spatial characteristic of the citrus canopy under various cultivation patterns. The citrus canopy under P1 was mostly an open-centered type with high canopy density in the middle layers (Figure 5a), which easily formed interlaminar closures in the canopy [31], resulting in reduced droplets penetration and spray coverage in the bottom layers [32,33]. On the contrary, citrus under P2 and P3 would have narrower canopy width and sparser foliage, which facilitated better spray

deposition in the middle layers and bottom layers. For P2, mean coverage on adaxial leaf surfaces in each layers were similar to that in P3. However, AAES resulted in a more uniform distribution in the canopy under P3 (CV between layers (adaxial) of 14.69%) than under P2 (CV between layers (adaxial) of 33.10%). It was possibly due to a more uniform canopy density under P3.

**Table 6.** Spray coverage for AAES under different cultivation patterns.

| Layer | Leaf Surface | Cultivation Pattern | | |
|---|---|---|---|---|
| | | General | Dense Dwarf | Dense Fence |
| Top | Adaxial | 11.91 | 9.42 | 15.69 |
| | Abaxial | 4.85 | 3.39 | 6.94 |
| Middle | Adaxial | 9.52 | 19.03 | 20.20 |
| | Abaxial | 2.88 | 12.56 | 11.92 |
| Bottom | Adaxial | 1.69 | 22.87 | 22.6 |
| | Abaxial | 1.68 | 7.03 | 9.29 |
| Mean spray coverage | Adaxial | 7.71 | 17.11 | 19.50 |
| | Abaxial | 3.14 | 7.66 | 9.38 |
| CV between layers | Adaxial | 56.64 | 33.10 | 14.69 |
| | Abaxial | 41.66 | 49.22 | 21.68 |

As shown in Table 6, mean spray coverage on abaxial leaf surfaces for AAES under three cultivation patterns were lower than that on adaxial leaf surfaces. Though air-assistance could aid in moving the foliage to increase spray coverage on abaxial leaf surfaces, the excessive canopy density under P1 still resulted in invalid coverage on abaxial leaf surfaces (mean coverage of 3.14%). On the other side, it was observed that the spray coverage on the abaxial leaf surfaces under P2 (mean coverage of 7.66%) and P3 (mean coverage of 9.38%) had significantly increased compared to P1. The spray coverage on the abaxial leaf surfaces under P3 (CV between layers (abaxial) of 21.68%) was more uniform than that under P2 (CV between layers (abaxial) of 49.22%).

In conclusion, AAES had the best spray performance in P3, followed by P2. The cultivation patterns significantly influenced the spray coverage and uniformity of spray distribution on leaf surfaces (adaxial and abaxial) in the canopy. As the canopy density decreased, the spray coverage and uniformity of spray distribution on leaf surfaces for AAES increased.

### 3.2.2. Comparison of Spray Coverage for Sprayers

Figure 6 shows the mean spray coverage on leaf surfaces (abaxial and adaxial) under three cultivation patterns for different sprayers. Though there were insignificant differences in the spray performances of ES and SG, both mean spray coverage on the adaxial leaf surfaces under three cultivation patterns showed similar orders: AAES > ES > SG. In comparison with AAES, the mean spray coverage on the adaxial leaf surfaces of ES and SG were reduced by 6.08% and 7.08%, respectively. For P1, AAES (7.71%) had slightly higher spray coverage on the adaxial leaf surfaces compared to ES (4.98%) and SG (5.56%) (Figure 6a). However, for P2 and P3, AAES (17.11% and 19.50%) had significantly higher spray coverage on the adaxial leaf surface compared to ES (9.37% and 11.37%) and SG (8.51 and 9.58%). On the one hand, due to the larger spray angle and spray distance (Table 2), the droplets discharged from AAES could cover more on the leaf surfaces at the same operating distance and forward speed. In addition, the air-charge-assistance ensured that the majority of droplets could reach on the target canopy. Meanwhile, the conventional sprayers had lower spray angle and spray distance without air-charge-assistance (Table 2), which caused weak performance in transporting droplets to the target canopy at the same operating distance and forward speed, resulting in a large amount of droplets deposited out of the canopy. However, this showed that, if the citrus had a high canopy density, the performance of AAES in improving the droplets penetration would be inadequate. To

achieve adequate spray coverage in citrus with a high canopy density, one of the radical solutions would be to increase the air flow velocity and spray pressure of AAES, which would make it prone to seriously drifting [12,34].

The results indicate that AAES had absolute higher spray coverage on the abaxial leaf surfaces under each pattern compared to ES as well as SG. Particularly under P2 and P3, the spray coverage on the abaxial leaf surfaces for AAES were 4.46% and 4.51% higher than that for ES and SG, respectively (Figure 6b). Such coverage trends were due to the sprayer operating method and droplet spectrum of nozzles used in the three sprayers. When the operator handled AAES to spray, the nozzle of AAES was located in front of the canopy, with the shaft of the nozzle forming an angle of 0–25° to the ground. This resulted in a better performance for airflow to increase canopy porosity, overturning the leaves so that the droplets can cover the abaxial leaf surface directly, improving the droplet deposition on the abaxial leaf surfaces [35]. In addition, AAES had a solid-cone nozzle, which could produce finer droplets. Such finer droplets under air–charge-assistance may have provided better droplets penetration and were helpful in increasing the spray coverage on abaxial leaf surfaces [10,36]. The operating method and nozzle of ES were the same as AAES; however, without air-charge-assistance, the ES had weak performance to move the spray to the abaxial leaf surface. Meanwhile, the nozzle of SG was located above the canopy during spraying, with the shaft of nozzle forming an angle of 60–75° to the ground. Such operating method may cause droplets to overlap on the adaxial leaf surfaces on the top layers. Moreover, the droplets emitted from SG were coarser than AAES, which further aggravated the overlap.

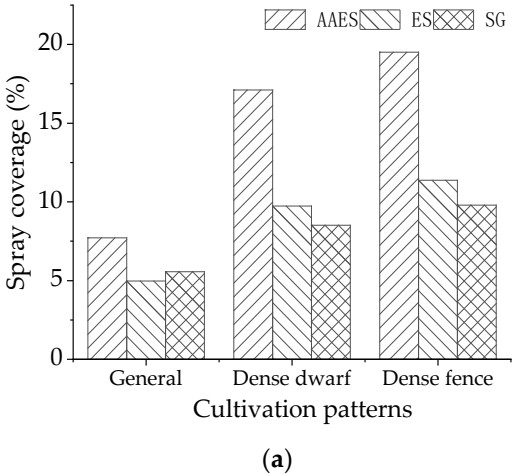
(**a**)

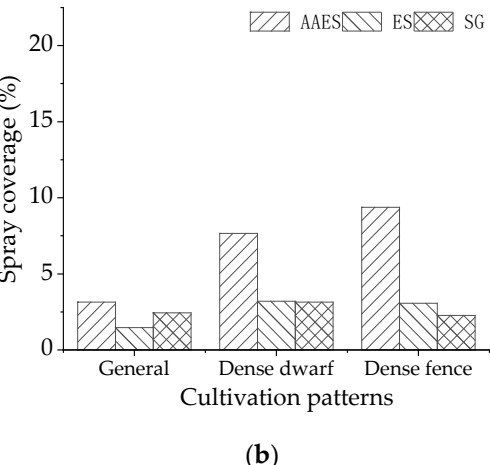
(**b**)

**Figure 6.** (**a**) Spray coverage on adaxial leaf surfaces for the tested sprayers; (**b**) Spray coverage on abaxial leaf surfaces for the tested sprayers. AAES: air-assisted electrostatic sprayer, ES: electric sprayer, SG: spray gun.

*3.3. Comprehensive Evaluation Model*

The equalization values of each comprehensive evaluation index after processing are shown in Table 7. The results of the equalization values indicate that the order of spray performance for the three sprayers were not completely consistent under different evaluation indexes. Such results were due to the lower values of the individual index ($I_4$ and $I_5$) compared to ES and SG. Therefore, it was necessary to construct a comprehensive evaluation model to analyze the spray performance of the sprayers.

Following the equalization values of the comprehensive evaluation index of the sprayers in Table 7, the comprehensive evaluation indexes and rank results are shown in Table 8. The results show that AAES obtained the highest comprehensive evaluation index for each cultivation pattern. This indicates that AAES had great spray performance while maintaining low resource consumption. Therefore, it could be concluded that AAES had the best comprehensive spray performance. In addition, the comprehensive evaluation

index calculated by the objective weighting method had a similar coefficient of variation (CV) to that calculated by the subjective weighting method: 28.71% and 27.34%, respectively. This indicates that the models constructed by different weight determination methods have great evaluation performance.

**Table 7.** Equalization value of each comprehensive evaluation index.

| Cultivation Pattern | Sprayer | $I_1$ | $I_2$ | $I_3$ | $I_4$ | $I_5$ |
|---|---|---|---|---|---|---|
| General | AAES | 1.80 | 1.27 | 1.34 | 0.98 | 0.58 |
| | ES | 0.76 | 0.82 | 0.62 | 1.03 | 2.08 |
| | BS | 0.44 | 0.91 | 1.04 | 0.98 | 0.34 |
| Dense dwarf | AAES | 1.80 | 1.45 | 1.64 | 0.84 | 0.85 |
| | ES | 0.76 | 0.83 | 0.69 | 0.69 | 0.89 |
| | BS | 0.44 | 0.72 | 0.67 | 1.46 | 1.25 |
| Dense fence | AAES | 1.80 | 1.44 | 1.91 | 1.14 | 1.51 |
| | ES | 0.76 | 0.84 | 0.63 | 0.41 | 0.65 |
| | BS | 0.44 | 0.72 | 0.46 | 1.45 | 0.84 |

$I_1$–$I_5$ respectively represent the equalization value of water consumption, mean coverage on adaxial leaf surfaces, mean coverage on abaxial leaf surfaces, CV between layers on adaxial leaf surfaces and CV between layers on abaxial leaf surfaces.

**Table 8.** Comprehensive evaluation index of each sprayer.

| Cultivation Pattern | Sprayer | SWM Evaluation Index | SWM Rank | OWM Evaluation Index | OWM Rank |
|---|---|---|---|---|---|
| General | AAES | 1.19 | 1 | 1.19 | 1 |
| | ES | 1.06 | 2 | 0.90 | 2 |
| | BS | 0.74 | 3 | 0.79 | 3 |
| Dense dwarf | AAES | 1.32 | 1 | 1.37 | 1 |
| | ES | 0.77 | 3 | 0.76 | 3 |
| | BS | 0.91 | 2 | 0.90 | 2 |
| Dense fence | AAES | 1.56 | 1 | 1.66 | 1 |
| | ES | 0.66 | 3 | 0.67 | 2 |
| | BS | 0.78 | 2 | 0.79 | 3 |

## 4. Conclusions

In this paper, the orthogonal experiment was used to optimize AAES. The spray performance of the optimized AAES was evaluated in a citrus orchard with various cultivation patterns compared to conventional sprayers. Lastly, comprehensive evaluation models were constructed to quantitatively evaluate the spray performance of the sprayers. The main conclusions are as follows.

The spray pressure factor had a significant influence on VMD and spray coverage. The air flow velocity had a significant influence on VMD. The applied voltage had a significant influence on CMR, but had insignificant influence on VMD and RS. This indicates that the influence of applied voltage on droplet atomization can be ignored when the spray pressure and air flow velocity are large enough [37]. The optimal spray parameters combination consisted of a spray pressure of 0.5 MPa, applied voltage of 9 kV and air flow velocity of 10 m/s.

The spray performance of AAES in P3 was better than that in the other patterns. The spray coverage on leaf surfaces (adaxial and abaxial) of AAES in the middle layer as well as the bottom layer under P1 were significantly lower than that under other patterns. Uniformity of droplets distribution was the best for P3 due to the sparse spatial construction within the canopy compared to the other patterns.

Comparing to the conventional sprayers, AAES had the highest mean spray coverage on leaf surfaces (adaxial and abaxial) due to the air-charge-assistance component. Though



some comprehensive indexes values were lower for AAES compared to ES and SG, the results of the comprehensive evaluation model constructed by both weight determination methods indicated that AAES had better spray performance in each cultivation pattern, with a lower resource consumption, while ensuring higher spray coverage on leaf surfaces as well as better uniform application.

**Author Contributions:** Conceptualization and methodology, X.X., K.Z. and N.L.; validation, X.X., Q.L. and Y.J.; formal analysis, Z.L., S.L. and S.S.; investigation, X.X. and K.Z.; data curation, X.X., K.Z. and N.L.; writing—original draft preparation, X.X.; writing—review and editing, K.Z., N.L. and Q.L.; funding acquisition, Z.L., S.L. and S.S. All authors have read and agreed to the published version of the manuscript.

**Funding:** This research was funded by the National Natural Science Foundation of China (31971797, 32271997), China Agriculture Research System of MOF and MARA (CARS-26), Key-Area Research and Development Program of Guangdong Province (2023B0202090001), General Program of Guangdong Natural Science Foundation (2021A1515010923), Special Projects for Key Fields of Colleges and Universities in Guangdong Province (2020ZDZX3061), Guangdong Provincial Special Fund For Modern Agriculture Industry Technology Innovation Teams (2023KJ108).

**Institutional Review Board Statement:** Not applicable.

**Data Availability Statement:** The data are available within the article.

**Acknowledgments:** The authors thank College of Electronic Engineering (College of Artificial Intelligence) of South China Agricultural University and National Citrus Industry Technical System Machinery Research Office for the facilities and support.

**Conflicts of Interest:** The authors declare no conflict of interest.

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
