# Peer review of "Parameters Optimization and Performance Evaluation Model of Air-Assisted Electrostatic Sprayer for Citrus Orchards"

_agriculture, doi:10.3390/agriculture13081498_

Round 1

Reviewer 1 Report

Please see the review .

Minor editing of English language required

Author Response

Dear reviewer,

Thank you very much for your opinions and suggestions. These comments are very helpful for revising and improving the quality of the manuscript. We have studied the opinions carefully and revised our manuscript, and further edited for English language. We response the reviewer’s comments with a point
by point and highlight the changes in the file. Please see the file for more details.

Reviewer 2 Report

The terminology has been abbreviated with letters, and in subsequent articles, only the letters need to be abbreviated. For example CMR, P1, P2, P3 etc.

In section 3.1 of the article, using range comparison and mean comparison to analyze the significance of the influence factors is a conventional method and not necessary to be explained. Additionally, this content belongs to the method section.

The author's statement is incorrect in line 248 of the article. The larger the CV, the more uneven the deposition.

In section 2.4 of the article, the author should explain the calculation method of evaluation index and the calculation method of OWM evaluation weight.

There is a question about Table 7, why are the flow rates of AAES and ES the same, but the values of water consumption I1 different?

About the quality of presentation, the English expressions needs to be improved, suggest finding a native English speaker to polish the article, so that it would be understood  easier.

Author Response

(The authors gave the same response as above.)
